# Antiplatelet Therapy during the First Year after Acute Coronary Syndrome in a Contemporary Italian Community of over 5 Million Subjects

**DOI:** 10.3390/jcm11164888

**Published:** 2022-08-20

**Authors:** Silvia Calabria, Felicita Andreotti, Giulia Ronconi, Letizia Dondi, Alice Campeggi, Carlo Piccinni, Antonella Pedrini, Immacolata Esposito, Alice Addesi, Nello Martini, Aldo Pietro Maggioni

**Affiliations:** 1Fondazione Ricerca e Salute (ReS)—Research and Health Foundation, 00187 Roma, Italy; 2Department of Cardiovascular Sciences, Fondazione Policlinico Universitario A. Gemelli IRCCS, 00168 Roma, Italy; 3Drugs and Health Srl., 00187 Roma, Italy; 4ANMCO Research Center, Heart Care Foundation, 50121 Firenze, Italy

**Keywords:** acute coronary syndrome, platelet aggregation inhibitors, health care costs, retrospective studies, treatment adherence, antiplatelet therapy

## Abstract

Background: Patterns of real-world antiplatelet therapy (APT) are reported to differ from guideline recommendations. This study describes patterns of APT during the year following a hospital diagnosis of acute coronary syndrome (ACS) and possible implications in terms of revascularization rates, rehospitalizations, and costs for the Italian National Health Service. Methods: From >5 million people, patients discharged (=index date) with primary/secondary ACS diagnosis in 2017 were identified by cross-linkage of administrative health data collected by the Ricerca e Salute (ReS) Foundation. Patients were characterized by revascularization rates at index date, APT at one month and one year (with appropriate coverage defined as ≥80% of defined daily doses), and rehospitalizations and healthcare costs during follow-up. Results: From the 2017 ReS database, 7966 (1.46 × 1000 inhabitants) were discharged alive with an ACS diagnosis. Most were >69 years and male. Of these, 83% (6640/7966) received ≥1 recommended antiplatelet agent within one month (treated group): 23% (1870/7966) as single and 60% (4770/7966) as dual APT. Among the 53% undergoing revascularization, 81% received dual APT at one month. Of the 78% with the same APT at one year, 66% showed appropriate coverage. For subjects treated and untreated with APT at one month, one-year rehospitalization rates were 54% and 66%, respectively, and mean per capita costs were EUR 14,316 and EUR 16,552, respectively (hospitalization driving >80% of costs). Conclusions: Among survivors of a hospitalized ACS diagnosis, this analysis shows relatively high APT under-treatment at one month and one year, associated with fewer index revascularization rates, more rehospitalizations, and greater costs. Further initiatives to understand undertreatment and poor adherence should lead to improved health management and savings.

## 1. Introduction

Antiplatelet therapy (APT) can prevent adverse outcomes in patients at risk of arterial thrombus formation, since the latter is largely driven by platelet activation and aggregation [1]. Acute coronary syndromes (ACS) are an expression of critical cardiac ischemia, often caused by sudden occlusive or sub occlusive thrombosis of a fissured atheromatous epicardial artery stenosis. Presentations range from cardiac arrest or hemodynamic instability (caused by malignant arrhythmias or mechanical complications) to more subtle manifestations [2]. Patients typically present acute chest discomfort and, according to the electrocardiogram, are differentiated into those with persistent ST-segment elevation (STE), whose mainstay of treatment is immediate revascularization by percutaneous coronary intervention (PCI), and those with no persistent ST-segment elevation (NSTE) [2]. The latter may include forms with non-obstructed epicardial coronary arteries, forms driven by tachyarrhythmia, anemia, hypertension, or sepsis, more than by epicardial artery obstruction, and forms such as unstable angina not accompanied by cardiac cell damage (typically assessed by a transient rise and fall of plasma cardiac troponin), for which revascularization may not have a role [2].

In patients with ACS, especially those undergoing PCI, a number of randomized trials have shown superior efficacy of dual (aspirin + P2Y12 platelet receptor inhibitor) compared to single (aspirin alone) APT [3,4,5] and of more potent compared to less potent P2Y12 inhibitors, albeit at a price of enhanced bleeding [6,7]. In parallel, in recent years, coronary stent technology has evolved toward less thrombogenic devices, while bleeding complications have emerged as harbingers of ominous prognosis [8], prompting the development of various APT strategies aimed at reducing bleeding risk [9,10].

A previous analysis of Italian administrative healthcare data showed that real-world prescriptions of APT differ from clinical practice guideline (CPG) recommendations [11]. The most recent CPGs recommend dual APT (DAPT) up to 12 months post-ACS (class of recommendation I, level of evidence A), preferably with aspirin + ticagrelor or aspirin + prasugrel over aspirin + clopidogrel [2]. For NSTE-ACS patients at high risk of bleeding, CPGs allow a shorter period of DAPT (dropping either the P2Y12 inhibitor or aspirin—class IIa, level A or B) or DAPT de-escalation (mainly switching to clopidogrel from a more potent P2Y12 inhibitor—class IIb, level A) [2].

Of note, the above CPG recommendations are based on efficacy/safety profiles of antiplatelet regimens tested in trials that selected patients at relatively low risk of drug-induced adverse reactions (namely bleeding). As a result, patients enrolled in such trials are typically younger and with fewer comorbidities compared to those in clinical practice. Administrative data, unlike clinical trials or registry studies, allow the analysis of unselected patient populations as well as independent and objective assessments of healthcare resource use and costs over time.

The present study conducted by the Fondazione ReS (Ricerca e Salute) investigates a large Italian National Health Service (INHS) population of ACS patients discharged in 2017, representative of the entire spectrum of ACS syndromes. Its aim is to investigate apparent discrepancies between guideline-recommended and real-world APT and the implications of such divergences for patients and healthcare systems. To this end, patient baseline characteristics, patterns of APT, rates of revascularization and rehospitalization, and overall healthcare resource use and costs during the year following discharge were analyzed.

## 2. Materials and Methods

### 2.1. Data Source

Under specific agreements, several local and regional Italian Health Authorities (IHAs) distributed throughout Italy authorized the Fondazione ReS to routinely collect their administrative data in the ReS database. The latter is physically located in the supercomputers of the non-profit Italian University consortium Cineca [12], which guarantees international quality and safety standards of data management. The INHS is a universal-coverage, single-stakeholder system; therefore, data collected by IHAs closely reflect the care of INHS beneficiaries. The analyzed data are the same transmitted by IHAs to the Italian Ministry of Health for reimbursement. The catchment community of over 5 million inhabitants shows an age distribution comparable to that reported for the entire country by the Italian Institute of Statistics (ISTAT) (Appendix A) [13]. Given the database representativeness, Fondazione ReS has conducted multiple observational studies for institutional purposes on a range of clinical questions [14,15,16].

### 2.2. Analyzed Data 

According to current European privacy legislation [17], the patient’s identification number was anonymized at the source. Different data categories were cross-linked by the ReS database. Demographic data included age, sex, IHA of residency, and disease waiver claim (i.e., co-payment). Pharmaceutical data included drugs reimbursed by the INHS and supplied by intra- and extra-hospital pharmacies, with defined daily dose (DDD) [18], number of packages, and dispensing date. Active drug substances were characterized by Italian marketing code and Anatomical Therapeutic Chemical code (World Health Organization’s ATC classification [18]). Hospitalization data included diagnoses and procedures reported on ordinary or day-hospital discharge forms, classified by the 9th International Classification of Disease Clinical Modification—ICD9CM [19] currently used in Italy. Outpatient data included specialist visits, tests, and procedures certified by the INHS. Direct costs paid by the INHS were recorded. Given data anonymization at the source and aggregated analyses for institutional purposes, the investigation is exempt from ethical approval and informed consent, in agreement with European legislation [17] and the involved IHAs.

### 2.3. Cohort Selection and Antiplatelet Regimens

In 2017 (accrual), ReS captured 5,469,430 INHS beneficiaries analyzable until the end of 2018 (~10% of the Italian resident population [13]). Those hospitalized and discharged at least once with a primary or secondary diagnosis of acute myocardial infarction—AMI (ICD9CM code 410.x, including all forms of AMI) or other acute and subacute forms of ischemic heart disease (ICD9CM code 411.x, including post-MI syndrome, intermediate coronary syndrome—i.e., unstable angina, and acute coronary occlusion without MI) were retained [19]. The most recent 2017 discharge corresponded to the index date. Patients were grouped by presence or absence of APT comprising CPG-recommended agents one month after the index date. Loss to follow-up was attributable to emigration from the area or to nursing home admission.

APT was categorized as single (SAPT) or dual (DAPT). SAPT (ATC V code) comprised aspirin (ASA, B01AC06), clopidogrel (B01AC04), ticlopidine (B01AC05), prasugrel (B01AC22), dipyridamole (B01AC07) or ticagrelor (B01AC24). DAPT comprised fixed-dose combinations (FDC) of dipyridamole/ASA or clopidogrel/ASA (B01AC30, distinguished by Italian marketing codes) or free combinations of antiplatelet agents. Only oral antiplatelet agents were analyzed.

### 2.4. Patient Clinical Characteristics and Index Revascularizations

Sex and age were assessed at index date. Arterial hypertension, dyslipidemia, chronic obstructive pulmonary disease (COPD) or asthma, diabetes, depression [20,21], and neoplasia were identified as comorbidities of interest at index date and within the previous year (see Appendix A for identification criteria).

Revascularizations during the index hospitalization, identified as PCI or coronary artery bypass graft surgery (CABG) through ICD9CM codes 36.x and 00.66, were analyzed by APT pattern.

### 2.5. Antiplatelet Treatment Coverage and Switching at One Year

APT coverage at follow-up was assessed by comparing mean annual doses to the DDD of each antiplatelet agent according to summaries of product characteristics. For patients who did not modify APT regimen during follow-up, treatment duration was considered appropriate when coverage was ≥80%. Among subjects who changed APT regimen during follow-up, the dispensation of other agents and the proportion of “switchers”, stratified by SAPT or DAPT category, were reported. During follow-up, more than one switch was possible for a given patient.

### 2.6. Other Antithrombotic Agents

The proportion of patients—treated or untreated with APT at one month—who received other antithrombotic agents (i.e., anticoagulants) within one month from the index date was assessed. Such agents comprised vitamin K antagonists—VKAs (ATC code B01AA), heparin (B01AF), pentasaccharide (B01AX05), and direct oral anticoagulants—DOACs (B01AF), including dabigatran (B01AE07).

### 2.7. Rehospitalizations and Healthcare Costs

Patients treated or untreated with APT at one month and readmitted to hospital with at least one ordinary or day-hospital stay during follow-up were analyzed. Causes of rehospitalization as reported on discharge forms were classified by first 10 primary ICD9CM main group diagnoses. Readmissions with a main/secondary diagnosis of severe bleeding (430–432 intracranial hemorrhage; 578.9 hemorrhages of gastrointestinal tract; 459.0 hemorrhages; 285.1 acute posthemorrhagic anemia; 280.0 chronic iron deficiency anemia secondary to blood loss) were also searched among patients receiving APT or not. 

Direct amounts paid by the INHS within one year after the index date were analyzed per patient and stratified by APT treatment or not. Since Italian administrative healthcare databases were created for reimbursement purposes, pharmaceutical costs can be extrapolated from prices of community and hospital pharmacies (inclusive of value-added tax). In-hospital costs were extrapolated from the DRG (Diagnosis-Related Group) classification. Costs of outpatient specialist care were based on current national tariffs. Integrated costs per patient were assessed by cross-linking different data sources.

### 2.8. Statistical Analyses

Given the observational nature of the analyses and the very broad catchment area, formal sample size was not calculated. Because very large numbers of patients/events originating from administrative data can result in conventional levels of statistical significance (two-sided *p* < 0.05) even for minimal differences of limited clinical significance, systematic *p* values were generally avoided, and nominal differences are reported. Continuous values are expressed as mean ± standard deviation (SD), and proportions as percentages. Analyses were performed using Oracle SQL Developer (Pasadena, CA, USA) Italian version 18.1.0.095.

## 3. Results

### 3.1. Cohort Selection and Antiplatelet Regimens

Of 5,469,430 inhabitants included in the ReS database in 2017, 8219 were hospitalized with a diagnosis of ACS (1.49 per 1000 inhabitants). Of these, 7966 (96.9%) were discharged alive (1.46 per 1000 inhabitants). Among them, 6790 (85.2%) received at least one antiplatelet drug within one month from discharge, while the remaining 1176 (14.8%) received no APT prescription. Among the 6790 receiving APT at one month, CPG-recommended agents were supplied to 4770 patients in the form of DAPT (70% of 6790 or 60% of 7966) and to 1870 patients in the form of SAPT (28% of 6790 or 23% of 7966), whereas 150 patients (2.2% of 6790 or 1.9% of 7966) received other APT. Given the very small size of the latter sample, this subgroup was not considered in further detail. The patients’ distribution by specific APT patterns is shown in Figure 1.

### 3.2. Clinical Characteristics

Males were prevalent in both groups of ACS patients (treated and untreated with APT at one month). Untreated patients were older—mean (±SD) 73 ± 12 vs. 69 ± 13 years—more often female, and with more comorbidities compared to treated ones (Table 1). Patients receiving clopidogrel only were also generally older, more often female, and with more comorbidities (excepting neoplasia) compared to patients receiving other APTs (Table 1). Patients receiving P2Y12 inhibitors other than clopidogrel (as SAPT or DAPT) were younger, less often female, and less often affected by COPD or depression compared to patients receiving other APTs (Table 1).

### 3.3. Revascularization Procedures

During the index hospitalization, 4228/7966 ACS patients discharged alive (53.1%) received either PCI or CABG. Of these, 3998 subjects (95%) were treated with APT at one month, and 3232 (81%) with DAPT. Of all patients receiving DAPT (4770), more than two-thirds (3232, i.e., 68%) underwent revascularization during the index hospitalization. Figure 2 compares the prevalence of overall APT and of DAPT in the whole population and in the revascularized group.

### 3.4. Annual Antiplatelet Treatment Coverage and Switching

Of the 6640 patients receiving guideline-recommended antiplatelet agents one month after the index date, 5161 (78%) remained on the same regimen at one year. Appropriate coverage (defined as ≥80% prescription of annual DDDs) was observed in 3418/5161 patients (66%) during follow-up (Table 2). Among non-switchers, the highest coverage was in those receiving ASA + prasugrel (79%) or ASA/clopidogrel as FDC (77%). 

Among the 3998 patients who received PCI/CABG during index hospitalization and at least one antiplatelet agent at one month, 3021 (76%) received the same agent at one year, and 2166 (72%) of these were considered appropriately covered.

Switching to a different antiplatelet regimen from the one prescribed at one month occurred in 1479/6640 (22%) subjects (Table 2). Most of those initially treated with ticagrelor alone (255/298; 86%) or prasugrel alone (65/75; 87%) switched to a different antiplatelet regimen. The most frequent switches were to ASA, clopidogrel, ASA/clopidogrel as FDC, or ticagrelor (Table 2).

### 3.5. Other Antithrombotic Agents

During the first month after the index date, 807/6790 patients treated with any APT (11.9%) and 253/1176 subjects not receiving APT (21.5%) were supplied with at least one other antithrombotic agent, namely parenteral or oral anticoagulants. Both treated and untreated APT groups received roughly equal proportions of heparins or pentasaccharide compared to DOACs or VKAs. Details are shown in Appendix A.

### 3.6. Rehospitalizations

During follow-up, 3637/6790 patients (53.6%) treated with APT versus 772/1176 patients (65.6%) untreated with APT at one month were readmitted to hospital (Table 3). 

Rehospitalizations were mostly related to CAD in APT-treated patients. Heart failure, endocarditis, and non-cardiac rehospitalizations were more frequent among untreated patients.

Of note, 1.9% of SAPT-treated patients (35/1870), 1.9% of DAPT-treated patients (89/4770), and 2.2% of untreated patients (26/1176) were readmitted within one year with a diagnosis of severe bleeding.

### 3.7. Healthcare Costs

The annual mean INHS cost per ACS patient treated or untreated with APT at one month was EUR 14,316 and EUR 16,552, respectively (Table 4). The annual mean expense per item of care is shown in Figure 3. Hospitalizations were the key driver of costs. In the group untreated with APT, the mean rehospitalization cost during follow-up surpassed that of the index hospitalization. In this group, the cost for non-cardiovascular drugs was greater than for cardiovascular ones (Table 4 and Figure 3).

## 4. Discussion

The present ReS analysis of almost 8000 unselected patients from a contemporary community of over five million inhabitants, discharged in 2017 from INHS hospitals with a diagnosis of ACS and followed for one year, identifies several issues. First, the baseline characteristics of the identified patients, both treated and untreated with APT at one month, show, on average, a higher mean age, a larger female portion, and a greater number of comorbidities (e.g., diabetes) compared to patients recruited in randomized clinical trials [3,4,5,6,7]. Other national administrative data studies have reported baseline characteristics comparable for age and sex to those in the present study [11,22,23], supporting the need for broader inclusion criteria in future randomized trials, in order to reliably transfer trial information to patients encountered in everyday practice.

Second, the antithrombotic regimens prescribed within one month from the index date show a broad range of patterns (Figure 1, Table 2 and Table 3). DAPT, as recommended by recent CPG [2], was prescribed to 4770/7966 (59.9%) patients. Among these, ASA + ticagrelor was the most prescribed (2258/4770, i.e., 47%), followed by ASA + clopidogrel in free combination (1177/4770, i.e., 25%) and ASA + clopidogrel as FDC (732/4770, i.e., 15%). The present 2017–2018 DAPT patterns differ from those reported for 2014–2015 when ASA + clopidogrel as FDC was the most prescribed (45%) followed by ASA + ticagrelor (31%) [11,23]. Thus, temporal trends suggest recent wider uptake of CPG-recommended therapies compared to previously.

Overall, a SAPT regimen at one month was prescribed to 1870/7966 (23%) of ACS patients. ASA was the most prescribed monotherapy (1055/1870, i.e., 56%) followed by clopidogrel (437/1870, i.e., 23%), ticagrelor (298/1870, i.e., 16%), and prasugrel (75/1870, i.e., 4%). Of note, 1176/7966 (15%) patients did not receive any APT prescription at one month, while 1909/7966 (25%) were prescribed a clopidogrel-based instead of a newer P2Y12 inhibitor-based DAPT, and 150/7966 (1.9%) received non-CPG-recommended agents. Thus, up to 64% of patients at one month (5105/7966) were apparently under-treated, i.e., receiving no APT, SAPT, clopidogrel-based DAPT, or other antiplatelet regimens. Each of these apparently under-treated groups—particularly the one on no APT—was older, with more frequent hypertension, diabetes, COPD, depression, and neoplasia, compared to patients prescribed DAPT involving a newer P2Y12 inhibitor (Table 1). The above suggests—at least in part—that caregivers may have adopted de-escalation strategies or refrained from APT on the basis of the patient’s high bleeding risk or frailty characteristics. Interestingly, recent trials dedicated to high bleeding risk groups undergoing PCI have shown net benefits of de-escalating new P2Y12 inhibitors to clopidogrel or of shortening DAPT to one month after PCI [24,25]. The more frequent rehospitalizations for severe bleeding and use of anticoagulation among patients receiving no APT at one month support the above interpretation.

As in previous analyses [11,23], APT in general and DAPT, in particular, were prescribed more often to patients undergoing revascularization during index hospitalization compared to the total population (95% vs. 85% and 81% vs. 70%, respectively, Figure 2). Conversely, only 41% of the non-revascularized population (1538/3738) received DAPT at one month (Figure 2). According to European Guidelines [2,26], patients with ACS due to epicardial artery-related STEMI or NSTEMI should undergo revascularization, if feasible, and receive DAPT for variable durations (usually up to 12 months) followed by SAPT, according to strategy and risk of bleeding. In this study, 46.9% of patients did not undergo revascularization during the index hospitalization, suggesting that prior revascularization, comorbidities, advanced age, extremely high bleeding risk, cases of MI with non-obstructive epicardial coronary arteries (MINOCA, where the benefit of DAPT needs more evidence), or late admission to community hospitals without revascularization facilities may have contributed to non-revascularization.

Anticoagulants were prescribed to 1060/7966 (13%) ACS patients at one month, consisting of DOACs in 395/1060 (37%) and VKAs in 275/1060 (26%). The rate of DOAC prescriptions increased compared to a similar 2014–2015 analysis (0.8%) [23], partly owing to DOAC reimbursement by the INHS in recent years. Patients untreated with APT at one month were more likely to receive anticoagulants compared to APT-treated patients (22% vs. 12%), suggesting that anticoagulation—by qualifying patients at high bleeding risk—overruled the prescription of APT. Indeed, anticoagulation could be one of the reasons for higher bleeding-related rehospitalization among untreated compared to APT-treated patients. 

Only two-thirds (66%) of patients treated with the same APT at 1 year received appropriate doses during follow-up (i.e., covering at least 80% of the year). Clopidogrel, its FDC with ASA, and the free association of ASA + prasugrel were the most persistently supplied regimens. While true patient self-administration cannot be assessed, drug supplies are considered reliable surrogates, allowing treatment coverage to be estimated. Most switchers receiving P2Y12 inhibitor monotherapy at one month changed to ASA during follow-up (Table 2).

Several factors may contribute to the observed APT patterns in a contemporary, unselected, real-world ACS population. These patterns likely reflect patient-related factors (age, side-effects, anticoagulation, other high bleeding risk comorbidities, depression) [11,27,28,29], prescriber-related factors (de-escalation strategies), and their combination (clinicians refraining from APT for highly comorbid, older, frail patients). Less common reasons may include the private purchase of medications (see below) or hospital provision of drugs at discharge [23]. Further understanding of apparent undertreatment in terms of patient characteristics and underlying caregiver reasons (contraindications, comorbidities, concomitant therapies) is warranted [11,23]. Whatever the causes, the patterns recorded by the present analysis are far from those in most trials or randomized registries, where patients are generally younger, less severe, and cared for by specialized cardiology units.

Of clinical and economic relevance is that over half of discharged ACS patients—both treated and untreated with APT at one month—were readmitted at least once during follow-up, mostly for CAD. The high readmission rate of patients treated with APT could be related to staged PCI procedures in multi-vessel disease, which generally happens within two months after the index discharge. 

As in previous analyses [11,23], rehospitalizations were more common among patients receiving no APT compared to APT-treated patients, attributable at least in part to the adverse consequences of undertreatment, with higher annual costs per capita (mean EUR 16,552 vs. EUR 14,316 in 2017–2018 and EUR 16,647 vs. EUR 13,297 in 2014–2015). Non-cardiovascular drugs were the main pharmaceutical cost among patients receiving no APT, supporting more complex clinical conditions. Among treated patients, the impact of cardiovascular drugs on the annual pharmaceutical burden was higher (58%) than in a similar 2014 analysis (34%) [11].

Strengths and Limitations

Because outpatient mortality is not available, survival at index discharge and INHS resource use within 31 December 2018 were used as proxies for permanence in the study. Data refer to patients cared for by the INHS, so private care was not recorded. According to a 2017 Italian Medicine Agency report [30], about 15% of drugs reimbursable by the INHS are purchased privately, especially less expensive ones (e.g., ASA), likely causing a small overestimation of untreated subjects. STE-MI was not distinguished from NSTE-MI, which have different revascularization rates [26]. The use of primary and secondary diagnoses and of high-sensitivity troponins may have led to the inclusion of ACS diagnoses secondary to hypertensive crises, supraventricular arrhythmias, pulmonary edema, or embolism, contributing to the 14.8% of patients receiving no APT at one month. The large sample and study design, however, reflect the entire spectrum of ACS syndromes and care in a contemporary community.

## 5. Conclusions

APT patterns of ACS patients in a real contemporary population and the related impact on national healthcare resource use and costs in the year following diagnosis were analyzed. A broad range of APT patterns, with high apparent under-treatment, was detected. Patients untreated with APT had more comorbidities and greater healthcare costs. The inclusion of less selected patients in future trials may contribute to defining guidelines more applicable to real populations. Further initiatives to understand the reasons for under-treatment and poor adherence may improve patient management and healthcare savings.

## Figures and Tables

**Figure 1 jcm-11-04888-f001:**
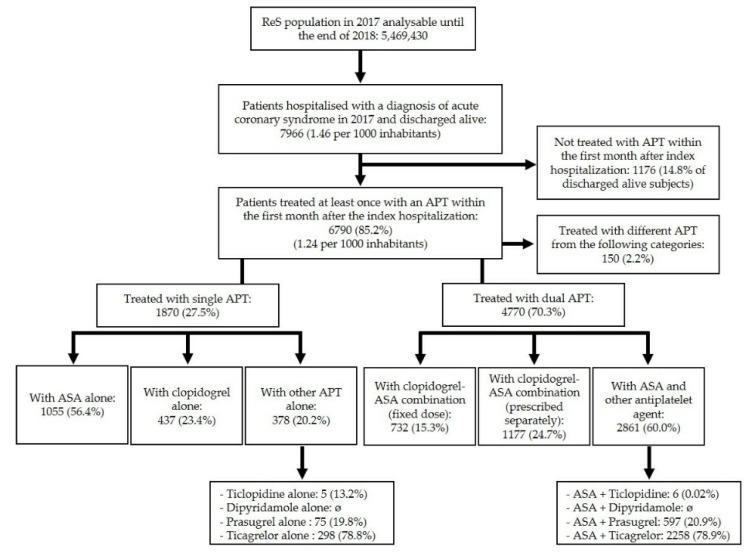
Patients discharged with acute coronary syndrome diagnosis in 2017 (index date) grouped by antiplatelet therapy (APT) or not at one month.

**Figure 2 jcm-11-04888-f002:**
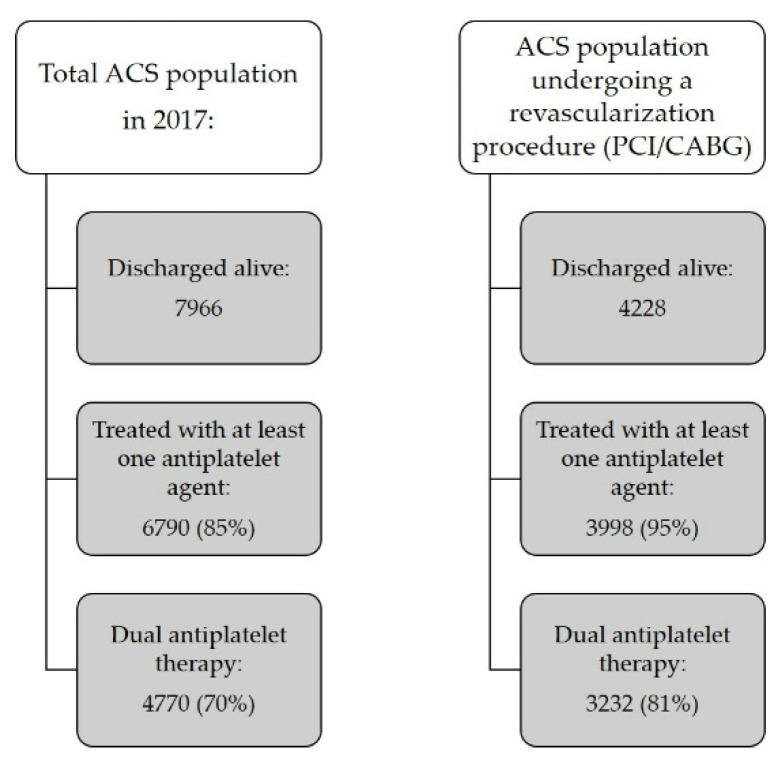
Prevalence of overall and dual antiplatelet therapy in the entire acute coronary syndrome (ACS) population and in those undergoing percutaneous (PCI) or surgical (CABG) revascularization.

**Figure 3 jcm-11-04888-f003:**
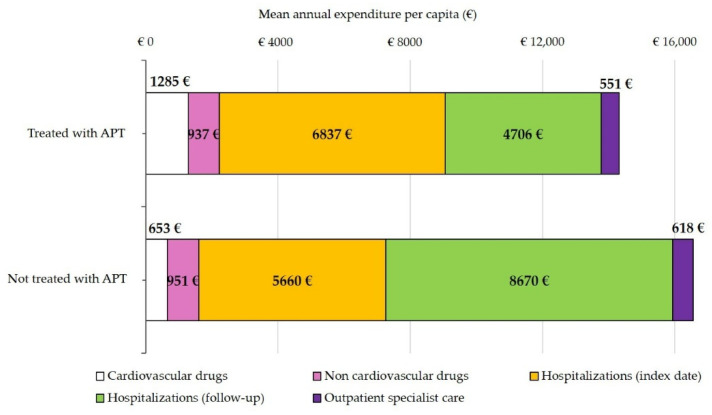
Mean annual per capita costs charged to the Italian National Health Service for acute coronary syndrome patients discharged in 2017, stratified by antiplatelet therapy (APT) at one month.

**Table 1 jcm-11-04888-t001:** Baseline characteristics of the acute coronary syndrome population stratified by type of antiplatelet therapy at one month.

	ACS Population, *n* = 7966							
	Patients with APT at 1 Month (*n* = 6790 *)	ASA Alone (*n* = 1055)	Clopidogrel Alone (*n* = 437)	Other APT Alone (*n* = 378)	Clopidogrel + ASA FDC (*n* = 732)	Clopidogrel + ASA not FDC (*n* = 1177)	ASA + Other APT (*n* = 2861)	Patients Untreated with APT (*n* = 1176)
Baseline demographic and clinical characteristics								
Mean age, years ± SD	69 ± 13	71 ± 12	75 ± 12	68 ± 11	74 ± 11	74 ± 12	64 ± 12	73 ± 12
Males, n (%)	4682 (69)	665 (63)	246 (56)	300 (79)	490 (67)	708 (60)	2183 (76)	684 (58)
Arterial hypertension	5225 (77)	872 (83)	400 (92)	323 (85)	626 (86)	1002 (85)	1873 (66)	972 (83)
Dyslipidemia	3292 (49)	532 (50)	276 (63)	238 (63)	428 (59)	601 (51)	1136 (40)	567 (48)
Diabetes	2200 (32)	369 (35)	178 (41)	150 (40)	296 (40)	415 (35)	736 (26)	435 (37)
COPD/Asthma	1050 (16)	194 (18)	103 (24)	47 (12)	171 (23)	245 (21)	256 (9)	290 (25)
Neoplasia	520 (8)	108 (10)	38 (9)	31 (8)	73 (10)	109 (9)	148 (5)	131 (11)
Depression	652 (10)	126 (12)	71 (16)	29 (8)	71 (10)	128 (11)	213 (7)	126 (11)

ACS: acute coronary syndrome; APT: antiplatelet therapy; ASA: aspirin; FDC: fixed dose combination; COPD: chronic obstructive pulmonary disease. * Including 150 patients receiving non-guideline recommended agents.

**Table 2 jcm-11-04888-t002:** Antiplatelet therapy regimens, appropriate coverage, and switching patterns during one-year follow-up in the acute coronary syndrome population.

Patients Treated with Guideline-Recommended Antiplatelet Agents at 1 Month after the Index Date, *n* = 6640										
Overall (*n* = 6640) *	ASA Alone (*n* = 1055)	Clopidogrel Alone (*n* = 437)	Ticlopidine Alone (*n* = 5)	Prasugrel Alone (*n* = 75)	Ticagrelor Alone (*n* = 298)	Clopidogrel + ASA FDC (*n* = 732)	ASA + Clopidogrel (*n* = 1177)	ASA + Ticlopidine (*n* = 6)	ASA + Prasugrel (*n* = 597)	ASA + Ticagrelor (*n* = 2258)
**Patients treated with the same APT during one year of follow-up, *n* (%)**										
5161 (78)	850 (81)	197 (45)	5 (100)	10 (13)	43 (14)	477 (65)	1006 (85)	4 (67)	553 (93)	2016 (89)
Patients with appropriate treatment coverage (≥80% of one-year follow-up dosing) among non-switchers, *n* (%)										
3418 (66)	569 (67)	93 (47)	2 (40)	7 (70)	29 (67)	366 (77)	473 (47)	0 (0)	437 (79)	1442 (72)
**Patients switching APT during follow-up, *n* (%)**										
1479 ^ (22)	From ASA alone (*n* = 205)	From Clopidogrel alone (*n* = 240)	From Ticlopidine alone (*n* = 0)	From Prasugrel alone (*n* = 65)	From Ticagrelor alone (*n* = 255)	From Clopidogrel + ASA FDC (*n* = 255)	From ASA + Clopidogrel (*n* = 171)	From ASA + Ticlopidine (*n* = 2)	From ASA + Prasugrel (*n* = 44)	From ASA + Ticagrelor (*n* = 242)
To ASA	-	227 (95)	-	63 (97)	245 (96)	111 (44)	-	-	-	-
To Clopidogrel	120 (59)	-	-	5 (8)	28 (11)	182 (71)	-	1 (50)	21 (48)	128 (53)
To Ticlopidine	5 (2)	-	-	-	-	1 (0)	6 (4)	-	-	3 (1)
To Prasugrel	6 (3)	-	-	-	4 (2)	2 (1)	1 (1)	-	-	29 (12)
To Ticagrelor	66 (32)	3 (1)	-	1 (2)	-	11 (4)	18 (11)	-	12 (27)	-
To ASA + Clopidogrel FDC	29 (14)	37 (15)	-	5 (8)	14 (6)	-	149 (87)	1 (50)	15 (34)	99 (41)

APT: antiplatelet therapy; ASA: aspirin; FDC: fixed dose combination. * Excluding 150 patients receiving non-guideline recommended agents. ^ Patients could have changed APT more than once; therefore, the total (1479) does not coincide with the sum of single items.

**Table 3 jcm-11-04888-t003:** Rehospitalizations during follow-up in the acute coronary syndrome population stratified by antiplatelet therapy at one month.

Cause of Rehospitalization		ACS Patients Treated withany APT at 1 Month (*n* = 6790)	ACS Patients Untreated with APT at 1 Month (*n* = 1176)
ICD9CM	Primary Diagnosis (Description)	Patients Rehospitalized within 1 Year, *n* (%)	
414	Coronary atherosclerosis, aneurysm, and dissection, other/unspecified form of chronic ischemic heart disease	1488 (21)	94 (8)
410	Acute myocardial infarction	914 (14)	131 (11)
411	Other acute/subacute form of ischemic heart disease	337 (5)	66 (6)
428	Heart failure	240 (4)	70 (6)
413	Chronic coronary syndrome	232 (3)	40 (3)
429	Undefined complication of heart disease	224 (3)	150 (13)
V43	Organ or tissue transplant	89 (1)	38 (3)
518	Lung disease	82 (1)	54 (7 5)
584	Acute renal failure	60 (1)	-
786	Other respiratory or chest symptoms	53 (1)	41 (3)
424	Endocarditis	-	25 (2)
	* Total number of rehospitalized patients at 1 year	3637 (54)	772 (66)

ACS: acute coronary syndrome; APT: antiplatelet therapy. * Because some patients were rehospitalized more than once and others for reasons not listed, totals do not coincide with sum of single items.

**Table 4 jcm-11-04888-t004:** Average per capita amounts paid by the Italian National Health Service during one-year follow-up for patients discharged in 2017 with a diagnosis of acute coronary syndrome.

Healthcare Category	ACS Population *n* = 7966	
	Treated with APT at One Month (*n* = 6790)	Untreated with APT at One Month (*n* = 1176)
	Mean Expense per Capita in € (% of Overall Expenditure; *% of Specific Category*)	
Drugs	2222 (16)	1604 (10)
*Cardiovascular*	*1285 (58)*	*653 (41)*
*Non-cardiovascular*	*937 (42)*	*951 (59)*
Hospitalizations	11,543 (81)	14,330 (87)
*Index date*	*6837 (59)*	*5660 (40)*
*During follow-up*	*4706 (41)*	*8670 (61)*
Outpatient specialist care	551 (4)	618 (4)
Annual total	14,316 (100)	16,552 (100)

ACS: acute coronary syndrome; APT: antiplatelet therapy.

## Data Availability

No new data were created or analyzed in this study. Data sharing is not applicable to this article.

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
