# Peer review of "Antiplatelet Therapy during the First Year after Acute Coronary Syndrome in a Contemporary Italian Community of over 5 Million Subjects"

_jcm, 2022, doi:10.3390/jcm11164888_

Round 1
Reviewer 1 Report
The authors present a study where they analyzed a large Italian National Health Service (INHS) population of ACS patients discharged in 2017. They aimed to investigate discrepancies between guideline-recommended and real-world APT and the implications of such discrepancies for patients and healthcare systems. The data were collected from > 5 million INHS beneficiaries of whom 8219 patients were hospitalized with ACS. Among them, 6790 (85.2%) received at least one antiplatelet drug within one month from discharge, while the remaining 1176 (14.8%) received no APT prescription. Among patients discharged with ACS 53.1% received either PCI or CABG. 95% of those patients received any APT, whereas 70% of those patients got DAPT. At follow-up (1 year), 78% of patients receiving APT remained on the same regimen. On the other hand, 53.6% treated with APT versus 65.6% untreated with APT at one month were readmitted to hospital. The annual mean INHS cost per ACS patient treated or untreated with APT at one month was €14,316 and €16,552, respectively. The authors concluded that Among survivors of a hospitalized ACS event, this analysis shows high APT undertreatment at one month and one year, associated with lower revascularization rates, more hospitalizations and greater costs. Further initiatives to understand undertreatment and poor adherence should lead to improved health management and savings. The current guidelines regarding management and treatment of patients following ACS gives high attention to duration of DAPT therapies. The authors analyzed enormous number of patients from their registry and should be congratulated for this. However I have several comment.
-Having access to such database based on ICD with high number of patients, why you did not analyze bleedings in population receiving DAPT vs APT vs no-APT? Patients are also hospitalized due to bleeding and during DAPT therapy they could also receive multiple transfusions which is also associated with increased cost.
-Patients untreated with APT were older, females and to have higher incidence of hypertension, diabetes COPD/Asthma, neoplasia, depression… in general with higher comorbidities thus more frail, suggesting that the medical team treating those patients, qualified them as high risk of bleeding patients, therefore those patients were more likely to have no APT after calculating the risk. Please comment.
-Of patients with ACS, 53.1% of the patients received a revascularization strategy (PCI or CABG). Of those, 95% received APT. How many patients receiving PCI or CABG were still on APT or DAPT at follow-up? On the other hand, Patients presenting with ACS due to coronary artery disease according to ESC Guidelines STEMI or NSTEMI, should have a revascularization strategy if feasible and APT or DAPT according to this strategy and risk of bleeding. In this study, 47% of the patients did not receive this strategy, suggesting that ACS was due to other factors such as MINOCA. The benefit of DAPT in MINOCA should be considered based on several conditions. However, the evidence is scarce. Therefore the use of DAPT in those 47% of patients is questionable unless it is well defined. Why to use DAPT in patients with no CAD who had ACS due to severe aortic stenosis?
-In patients with APT, 41% of the patients were readmitted due to CAD,aneurysm and dissection, other unspecified form of chronic ischemic heart disease were as only 12% in the no -APT. 25% were readmitted in the APT group due to MI vs 17% in the no APT group, 9% vs 8% due to other acute/subacute ischemic heart disease. Please comment on this, is this high readmission rate in the APT group related to staged PCI procedure in MVD? And why you concluded that patients with APT had lower revascularization rate?
-patients in the no APT group had higher rates of organ or tissue transplant at follow-up, higher rates of readmission due to lung disease (Lung disease + other respiratory or chest symptoms) as well as higher incidence of readmission due to endocarditis. Those patients are known to have longer hospital stay as well as high treatment related cost. Thus the cost related analyses could be biased.
- The authors should reconsider their methodology, I think It would be of benefit to aim their study on patients strictly with CAD and those who got revascularization strategy and relate it with current RCTs and guidelines.
Author Response
REVIEWER #1
The authors present a study where they analyzed a large Italian National Health Service (INHS) population of ACS patients discharged in 2017. They aimed to investigate discrepancies between guideline-recommended and real-world APT and the implications of such discrepancies for patients and healthcare systems. The data were collected from > 5 million INHS beneficiaries of whom 8219 patients were hospitalized with ACS. Among them, 6790 (85.2%) received at least one antiplatelet drug within one month from discharge, while the remaining 1176 (14.8%) received no APT prescription. Among patients discharged with ACS 53.1% received either PCI or CABG. 95% of those patients received any APT, whereas 70% of those patients got DAPT. At follow-up (1 year), 78% of patients receiving APT remained on the same regimen. On the other hand, 53.6% treated with APT versus 65.6% untreated with APT at one month were readmitted to hospital. The annual mean INHS cost per ACS patient treated or untreated with APT at one month was €14,316 and €16,552, respectively. The authors concluded that Among survivors of a hospitalized ACS event, this analysis shows high APT undertreatment at one month and one year, associated with lower revascularization rates, more hospitalizations and greater costs. Further initiatives to understand undertreatment and poor adherence should lead to improved health management and savings. The current guidelines regarding management and treatment of patients following ACS gives high attention to duration of DAPT therapies. The authors analyzed enormous number of patients from their registry and should be congratulated for this. However I have several comment.
-Having access to such database based on ICD with high number of patients, why you did not analyze bleedings in population receiving DAPT vs APT vs no-APT? Patients are also hospitalized due to bleeding and during DAPT therapy they could also receive multiple transfusions which is also associated with increased cost.
RESPONSE: We thank the Reviewer for raising this question. We fully agree that this would increase the value of our work. Therefore, we have searched for hospitalizations (ICD-9CM codes) related to a diagnosis of severe bleeding (intracranial hemorrhage, gastrointestinal tract hemorrhage, hemorrhage, acute post hemorrhagic anemia, chronic iron deficiency anemia secondary to blood loss) within the year of follow-up. This analysis shows that untreated patients were more frequently readmitted with a diagnosis of severe bleeding (2.2%) compared to patients receiving APT (1.9%). The following sections have been updated accordingly:
- Methods: “Readmissions with a main/secondary diagnosis of severe bleeding (430-432 intracranial hemorrhage; 578.9 hemorrhage of gastrointestinal tract; 459.0 hemorrhage; 285.1 acute post hemorrhagic anemia; 280.0 chronic iron deficiency anemia secondary to blood loss) were also searched among patients receiving APT or no.”;
- Results: ”Of note, 1.9% of SAPT-treated patients (35/1870), 1.9% of DAPT-treated patients (89/4770) and 2.2% of untreated patients (26/1176) were readmitted within one year with a diagnosis of severe bleeding.”;
- Discussion: “Patients untreated with APT at one month were more likely to receive anticoagulants compared to APT-treated patients (23% vs 13%), suggesting that anticoagulation - by qualifying patients at high bleeding risk - overruled the prescription of APT. Indeed, anticoagulation could be one of the reasons for higher bleeding-related re-hospitalization among untreated compared to APT-treated patients.”.
-Patients untreated with APT were older, females and to have higher incidence of hypertension, diabetes COPD/Asthma, neoplasia, depression… in general with higher comorbidities thus more frail, suggesting that the medical team treating those patients, qualified them as high risk of bleeding patients, therefore those patients were more likely to have no APT after calculating the risk. Please comment.
RESPONSE: We thank the Reviewer for this suggestion. The Discussion section has been updated in several points as follows: “The above suggests - at least in part - that caregivers may have adopted de-escalation strategies or refrained from APT on the basis of patient high bleeding risk or frailty characteristics”. “Patients untreated with APT at one month were more likely to receive anticoagulants compared to APT-treated patients (23% vs 13%), suggesting that anticoagulation - by qualifying patients at high bleeding risk - overruled the prescription of APT.” “These patterns likely reflect patient-related factors (advanced age, treatment side-effects, indications for oral anticoagulation, other high bleeding risk comorbidities, depression) [11,27-29], or prescriber-related factors (de-escalation strategies), and their combination (clinicians refraining from APT for highly comorbid, older, frail patients).”
-Of patients with ACS, 53.1% of the patients received a revascularization strategy (PCI or CABG). Of those, 95% received APT. How many patients receiving PCI or CABG were still on APT or DAPT at follow-up? On the other hand, Patients presenting with ACS due to coronary artery disease according to ESC Guidelines STEMI or NSTEMI, should have a revascularization strategy if feasible and APT or DAPT according to this strategy and risk of bleeding. In this study, 47% of the patients did not receive this strategy, suggesting that ACS was due to other factors such as MINOCA. The benefit of DAPT in MINOCA should be considered based on several conditions. However, the evidence is scarce. Therefore the use of DAPT in those 47% of patients is questionable unless it is well defined. Why to use DAPT in patients with no CAD who had ACS due to severe aortic stenosis?
RESPONSE: We thank the Reviewer for the suggestions. Among patients who had undergone PCI/CABG during index hospitalization and were treated with at least one antiplatelet agent at one month, we have calculated the annual coverage based on total DDDs. The Results section has been updated: “Among the 3998 patients who received PCI/CABG during index hospitalization and at least one antiplatelet agent at one month, 3021 (76%) received the same agent at one year, and 2166 (72%) of these were considered appropriately covered.”
Moreover, in line with the Reviewer’s comments, the Discussion section has been updated as follows:
“According to European Guidelines [2,26], patients with ACS due to epicardial artery-related STE-MI or NSTE-MI should undergo revascularization, if feasible, and receive DAPT for variable durations (usually up to 12 months) followed by SAPT, according to strategy and risk of bleeding. In this study, 46.9% of patients did not undergo revascularization during index hospitalization, suggesting that prior revascularization, comorbidities, advanced age, extremely high bleeding risk, cases of MI with non-obstructive coronary arteries (MINOCA, where benefit of DAPT needs more evidence), or late admission to community hospitals without revascularization facilities may have contributed to non-revascularization.”
-In patients with APT, 41% of the patients were readmitted due to CAD, aneurysm and dissection, other unspecified form of chronic ischemic heart disease were as only 12% in the no -APT. 25% were readmitted in the APT group due to MI vs 17% in the no APT group, 9% vs 8% due to other acute/subacute ischemic heart disease. Please comment on this, is this high readmission rate in the APT group related to staged PCI procedure in MVD? And why you concluded that patients with APT had lower revascularization rate?
RESPONSE: Thank you for these comments which have led us to substitute proportions of patients with a specific diagnosis on the total of treated and untreated cohorts with proportions on the total of hospitalized treated (N=3637) and untreated (N=772) patients in Table 3. We comment in the Results as follows: “Re-hospitalizations were mostly limited to CAD in APT-treated patients. Heart failure, endocarditis and non-cardiac re-hospitalizations, instead, were more frequent among untreated patients.”
The high readmission rate of patients treated with APT could be related to staged PCI procedure in multi-vessel disease, that generally happens within two months after the index discharge. The same sentence has been added in the Discussion section.
The revascularization procedures we report are those during the index hospitalization (not during follow-up). In the Abstract-conclusions and in the Discussion we comment as follows:
- Abstract Conclusions: “Among survivors of a hospitalized ACS diagnosis, this analysis shows relatively high APT under-treatment at one month and one year, associated with fewer index revascularizations rates, more re-hospitalizations and greater costs.”;
- Discussion: “In this study, 46.9% of patients did not undergo revascularization during index hospi-talization, suggesting that prior revascularization, comorbidities, advanced age, extremely high bleeding risk, cases of MI with non-obstructive coronary arteries (MINOCA, where benefit of DAPT needs more evidence), or late admission to community hospitals without revascularization facilities may have contributed to non-revascularization.”.
-patients in the no APT group had higher rates of organ or tissue transplant at follow-up, higher rates of readmission due to lung disease (Lung disease + other respiratory or chest symptoms) as well as higher incidence of readmission due to endocarditis. Those patients are known to have longer hospital stay as well as high treatment related cost. Thus the cost related analyses could be biased.
RESPONSE: We thank the Reviewer for this discussion point. Since the aim of the present study, based on administrative data, was to take a real-world snap-shot of patients with ACS, the fact that those untreated with APT are more frail, have more re-hospitalizations and longer in-hospital stay does not bias the cost findings in any way. The higher the clinical complexity of patients, the higher the healthcare and economic burden. Our study clearly shows this equation, thanks to the objective inclusion of an unselected population.
- The authors should reconsider their methodology, I think It would be of benefit to aim their study on patients strictly with CAD and those who got revascularization strategy and relate it with current RCTs and guidelines.
RESPONSE: We thank the Reviewer for the suggestion of performing an analysis strictly focused on patients with CAD who underwent a revascularization procedure. Actually, all patients included in this analysis were discharged with a diagnosis of ACS with or without a revascularization procedure at the index hospitalization, and with or without APT. In this sense, we followed exactly the methodology suggested by the Reviewer.
The comparison with current RCTs and guidelines has already been addressed in the Discussion section. A major asset of our investigation is to highlight the high level of clinical and treatment complexity in real world practice that explains at least in part the apparent discrepancies from RCT data guideline recommendations.

Reviewer 2 Report
Dear Authors, I have read your manuscript with interest.
The current manuscript titled: "Antiplatelet therapy during the first year after acute coronary syndrome in a contemporary Italian community of over 5 million subjects" represents an important analysis of evolving field of Cardiology.
The title reflects the manuscript content and helps the reader navigate the article essence.
In my opinion, these are the adjustments which should be made to increase the value of your manuscript:
1. Please, change the manuscript title words to capital letters.
2. In Abstract, line 31, change please “yrs” to “years”.
3. In keywords, add please “Antiplatelet therapy”.
4. In Introduction chapter, add please definition and detailed general information about Acute Coronary Syndrome and its forms.
5. Lines 111-112: specify please what include “acute myocardial infarction” and “other acute and subacute form of ischaemic heart disease”.
6. Line 124-126, chapter 2.4: add please definition characteristics for arterial hypertension, dyslipidaemia and diabetes.
7. Please specify how the neoplasia were diagnosed, whether you participated in this process or whether it was an established diagnosis in the anamnesis.
8. Please change the Figures Font to Palatino.
9. Why depression was included in this study, explain please its signification.
10. In Results chapter, add please Acute Coronary Syndrome forms and its particularities.
11. Lines 229-230: delete please abbreviations and add them under the Table 3.
12. I advise to change “angina pectoris” to “Chronic coronary syndrome”.
13. Change please “i.e.” to “i.g.,” – lines 253, 261, 262, 268, 326, etc.
14. Lines 301-303: explain contribution mechanisms of patients-related factors, especially the depression role.
15. The manuscript contains some punctuation errors, please revise the text.
Author Response
REVIEWER #2
Dear Authors, I have read your manuscript with interest.
The current manuscript titled: "Antiplatelet therapy during the first year after acute coronary syndrome in a contemporary Italian community of over 5 million subjects" represents an important analysis of evolving field of Cardiology.
The title reflects the manuscript content and helps the reader navigate the article essence.
In my opinion, these are the adjustments which should be made to increase the value of your manuscript:
- Please,change the manuscript title words to capital letters.
Ok, the title has been changed.
- In Abstract, line 31, change please “yrs” to “years”.
Ok, it has been changed.
- In keywords, add please “Antiplatelet therapy”.
Ok, it has been added.
- In Introduction chapter, add please definition and detailed general information about Acute Coronary Syndrome and its forms.
We thank the Reviewer for the suggestion. A brief description of ACS and its forms has been given in the Introduction: “Acute coronary syndromes (ACS) are an expression of critical cardiac ischemia, often caused by sudden occlusive or sub occlusive thrombosis of a fissured atheromatous epicardial artery stenosis. Presentations range from cardiac arrest or hemodynamic in-stability (caused by malignant arrhythmias or mechanical complications) to more subtle manifestations [2]. Patients typically present acute chest discomfort and, according to the electrocardiogram, are differentiated into those with persistent ST-segment elevation (STE), whose mainstay of treatment is immediate revascularization by percutaneous coronary intervention (PCI), and those with no persistent ST-segment elevation (NSTE) [2]. The latter may include forms with non-obstructed epicardial coronary arteries, forms driven by tachyarrhythmia, anemia, hypertension, or sepsis, more than by epicardial artery obstruction, and forms such as unstable angina not accompanied by cardiac cell damage (typically assessed by a transient rise and fall of plasma cardiac troponin), for which revascularization may not have a role [2].”
- Lines 111-112: specify please what include “acute myocardial infarction” and “other acute and subacute form of ischaemic heart disease”.
These wordings describe the 2007 version of the ICD9-CM codes currently used to identify diagnoses and procedures in the discharge forms of Italian National Health Service-affiliated hospitals. Code 410.x includes all forms of acute myocardial infarction, while code 411.x includes post-myocardial infarction syndrome and unstable angina. Clarified descriptions, relative reference and related link have been updated in section 2.3.
- Line 124-126, chapter 2.4: add please definition characteristics for arterial hypertension, dyslipidaemia and diabetes.
The specific definitions are provided in Table S1 of the supplementary materials.
- Please specify how the neoplasiawere diagnosed, whether you participated in this process or whether it was an established diagnosis in the anamnesis.
The identification criteria for neoplasia are listed in Table S1 of the supplementary material. Administrative databases collect every healthcare service use provided by the INHS in an electronic way, whereas individualized clinical data are unavailable. Thus, the criteria for disease identification (e.g. neoplasia) or chronic medical conditions (e.g. COPD) are retrieved by specific classification systems or tariffs codes listed in section 2.2.
- Please change the Figures Font to Palatino.
The figure fonts have been changed to Palatino.
- Why depression was included in this study, explain please its signification.
The choice to include depression among the comorbidities can be explained by its frequent presence among real-world patients analyzed by this study, who are older than those recruited in clinical trials and are frequently treated with antidepressants, mainly in a chronic manner without careful recognition or quality control for appropriateness and duration. Patients with depression are generally excluded from trials, because they are considered less reliable in terms of compliance with study procedures. Moreover, depression is generally considered a cause of poor adherence, therefore we think that this is an important concomitant disease to be considered when prescription patterns are assessed. Finally, studies on the effects of antidepressants on cardiovascular events in patients with major depression and cardiovascular disease produced conflicting results, with some population-based studies showing an increased risk of major adverse cardiovascular events. Thus, it is important, in our opinion, to study the frequency of this clinical condition in patients with ACS in real life settings. The analysis of some comorbidities of interest is important, given the real-life picture taken by this type of study, as opposed to clinical trials. The text dealing with depression has been enriched with some references.
- In Results chapter, add please Acute Coronary Syndrome forms and its particularities.
Thank you for this comment. Through administrative data, different forms of ACS cannot be analyzed, given the absence of clinical data that guarantee the diagnosis. On the other hand, we thank the Reviewer for allowing us to specify - in the Introduction and in the Strengths and Limitations sections – that the present study is “representative of the entire spectrum of ACS syndromes.”
- Lines 229-230: delete please abbreviations and add them under the Table 3.
Ok, legends have been added to the tables to explain abbreviations.
- I advise to change “angina pectoris” to “Chronic coronary syndrome”.
We thank the Reviewer for the suggestion. The change has been made.
- Change please “i.e.” to “i.g.,” – lines 253, 261, 262, 268, 326, etc.
Thank you, however, “i.e.” has been used as a substitute of “that is”, while “e.g.” has been used as a substitute of “for example”.
- Lines 301-303: explain contribution mechanisms of patients-related factors, especially the depression role.
We thank the Reviewer for allowing us to explain this sentence. The study observed an undertreatment of APT that was probably due to some known factors influencing drug compliance, in general, both patient-related and prescriber-related factors. The sentence has been reinforced by some references.
- The manuscript contains some punctuation errors, please revise the text.
Ok, the text has been revised.

Round 2
Reviewer 1 Report
The authors had addressed all my comments. I have no further comments.
Reviewer 2 Report
I agree with the changes made, which significantly improve the quality of the manuscript. I recommend this article for publication. Good luck!